# An Overview on Assay Methods to Quantify ROS and Enzymatic Antioxidants in Erythrocytes and Spermatozoa of Small Domestic Ruminants

**DOI:** 10.3390/ani13142300

**Published:** 2023-07-13

**Authors:** Valeria Pasciu, Maria Nieddu, Francesca Daniela Sotgiu, Elena Baralla, Fiammetta Berlinguer

**Affiliations:** 1Department of Veterinary Medicine, University of Sassari, 07100 Sassari, Italy; fdsotgiu@uniss.it (F.D.S.); ebaralla@uniss.it (E.B.); berling@uniss.it (F.B.); 2Department of Medicine, Surgery and Pharmacy, University of Sassari, 07100 Sassari, Italy; marvi@uniss.it

**Keywords:** ROS, enzymatic antioxidant, spermatozoa, erythrocytes, small domestic ruminants, method assay

## Abstract

**Simple Summary:**

The oxidative stress in the cell is the result of an imbalance between oxidants and the antioxidant defense system. In particular, oxidative stress in erythrocytes and spermatozoa of small domestic ruminants can negatively affect their productivity and welfare. This review highlights the assay methods for the quantification of ROS and enzymatic antioxidant activity in these cells, in order to help the researchers to evaluate the intracellular oxidative status and antioxidant defenses. These data can be useful in monitoring the health and reproduction of small domestic ruminants.

**Abstract:**

The present review aims to provide an overview of the assay methods for the quantification of ROS and principal enzymatic antioxidants as biomarkers of oxidative stress in erythrocytes and spermatozoa of small domestic ruminants. A complete literature search was carried out in PubMed, Scopus and the World Wide Web using relevant keywords and focusing on the last five years (2018–2023). Among spectrophotometry, fluorometry and chemiluminescence, the most widely used method for ROS assay is fluorometry, probably because it allows to simultaneously assay several ROS, using different probes, with greater economic advantages. Regarding intracellular antioxidant enzymes, recent literature reports only spectrophotometric methods, many of which use commercial kits. The use of a less sensitive but cheapest method is suitable because both erythrocytes and spermatozoa samples are highly concentrated in domestic ruminant species. All methods considered in this review have been found to be appropriate; in general, the differences are related to their costs and sensitivity. Quantification of ROS and enzymatic antioxidant activity in erythrocytes and spermatozoa may find application in the study of the welfare and health status of small domestic ruminants for monitoring livestock production.

## 1. Introduction

Oxygen is a fundamental element in aerobic life and oxidative metabolism representing the principal energy source for aerobic cells [1,2]. Reactive oxygen species (ROS) are generated by a variety of cellular metabolic activities and as a by-product of ATP generation mediated by mitochondrial respiration [3,4]. ROS are engaged in many redox-governing cell activities for the preservation of cellular homeostasis [5]. Redox balance is the basis of animal welfare and health; oxidative stress (OS) in the cell is the result of an imbalance between the level of ROS and the antioxidant defense systems [6]. A decrease in antioxidants defenses or an increase in oxidants level causes an imbalance in favor of the oxidative state, which could have harmful effects, such as the development of tumors [7,8,9], reproductive problems [10,11] and conditions altering animal health and welfare [12,13]. Figure 1 summarizes the imbalance between antioxidants and oxidants in the cell and the consequences of OS. Nowadays, increasing research efforts focusing on the study of OS have proven that it is linked to animal and human nutrition, health, reproduction, growth and environmental pollution [10,11,12,13,14]. Furthermore, environmental contaminants (bisphenols, pesticides, metals and metalloids) have also been gaining more attention in the livestock sector because of their harmful effects on animals’ productivity and fertility, which is associated with the production of ROS [15,16].

ROS and other oxidants can cause the oxidation of DNA, proteins and lipids with consequent membranes and tissue damage that can lead to cell death via apoptosis or necrosis [17,18]. In Table 1, the principal ROS, along with the metabolic way from which they derive, are summarized.

Cells’ redox balance is maintained by the action of antioxidant enzymes, such as superoxide dismutase (SOD), catalase (CAT), glutathione peroxidase (GPX), glutathione reductase (GSR) and other substances, such as glutathione (GSH), total thiols, vitamins E, C and A, that reduce the excess of ROS [25]. In Table 2, the main enzymatic antioxidants with their physiological functions are reported.

The overall oxidative state of the cell, with the main ROS and the system of enzymatic antioxidants, are summarized in Figure 2.

Recently, several studies have focused on the evaluation of OS in animals in order to provide useful information to farmers for improving animal’s health and welfare and obtain a better production of meat or milk by addition of antioxidants or rebalancing diet. Song et al. [30] reported that reduced milk production in cows with ketosis is partly due to increased OS along with mitochondrial dysfunction in the mammary gland.

A study by Zheng et al. [31] showed that cows with high serum ROS content may suffer oxidative damage more than those with low ROS levels. In these animals, the levels of several antioxidants, including CAT, GPX and SOD, increased proportionally to ROS. Higher serum activity of CAT, GPX and SOD may be a physiological response to neutralize/mitigate the action of ROS [31].

Majrashi et al. [32] reported that aging can induce OS and decrease ATP production and mitochondrial function in goat testis. Antioxidants that are naturally present in cells and biological fluids fail to eliminate the excessive amount of ROS produced during aging, leading to cell death and tissue damage. Therefore, the addition of antioxidants in feed could significantly reduce OS and improve mitochondrial function, resulting in improved goat health.

Çelik et al. [33] reported that OS in sheep, caused by a change in altitude, could be reduced by supplementing the feed with antioxidants. This appears to prevent OS from negatively affecting meat quality in sheep.

Different authors indicated a relationship between heat stress (HS) and OS, due to similar genes expressed after heat or oxidant agents’ exposure [34,35]. Indeed, HS was also proven to increase antioxidant enzyme activities (SOD, CAT and GPX) as a consequence of increased ROS levels due to heat exposure [36].

HS was proven to be responsible of inducing oxidative stress in sheep [36,37,38]. The increased environmental temperature and humidity that occur during summer can compromise animal production in livestock industries [39].

Slimen et al. [34] reported that HS affects the oxidative status of blood sheep leading to an overproduction of transition metal ions, that determine electron donations to oxygen, and formation of superoxide anion and/or hydrogen peroxide.

Furthermore, Shahat et al. [40] demonstrated that mild HS induces deleterious effects on animal health and reproduction, causing total and/or progressive reduction in sperm motility and acrosome integrity of fresh spermatozoa. These effects can be mitigated by slow release of substances with antioxidant effect as melatonin, taken before HS exposure.

The importance of studying OS for the assessment of animal welfare and health has led researchers to focus on studying the imbalance between oxidants and antioxidants in animal cells. Specifically, in this review, we focused on the study of two particular mammalian cells, erythrocytes and spermatozoa. These cells are unable to activate transcription processes in order to increase their antioxidant defenses, because of their intrinsic nature. Erythrocyte is, in fact, an enucleated cell in mammalian and the spermatozoon is a haploid cell.

Spermatozoa and erythrocytes are highly susceptible to the deleterious effects of ROS due to the large amount of unsaturated fatty acids found in their cell membranes. ROS promote peroxidation of lipids, resulting in loss of membrane integrity with a consequent increase in permeability. Malondialdehyde (MDA), a typical product of lipid peroxidation by ROS, can crosslink phospholipids and proteins and oxidize sulfhydryl groups of proteins, damaging the cell membrane and causing hemolysis [18,41]. A group of researchers found that the addition in vitro of pro-oxidant molecules to sheep erythrocytes led to an increase in MDA and ROS levels and damage the erythrocyte membranes [18]. In erythrocytes, high levels of ROS can determine eryptosis [42], and the addition of antioxidants protects them from this fatal event [43].

Also, spermatozoa are very vulnerable to oxidative attack because their cellular membrane contains an elevated quantity of polyunsaturated fatty acids (PUFA), highly sensitive to lipid peroxidation. Furthermore, these cells contain a large number of mitochondria, that provide the necessary energy for sperm motility, but they constitute also the major source of ROS in spermatozoa [44].

Membrane integrity is critical for both erythrocytes and spermatozoa. Indeed, the function of erythrocytes is to promote gas exchange through the small capillary circulation [45,46], while for spermatozoa, membrane integrity is essential to maintain their fertilizing capacity and to undergo the acrosomal reaction [27].

OS in spermatozoa is linked to male infertility, reduced sperm motility, sperm DNA damage and increased risk of recurrent abortions and genetic diseases [27,47]. Sperm motility is essential for normal fertilization, and the orientation of motility and the motor strategy of spermatozoa are crucial features as well. Consequently, OS impairs the main functionality of spermatozoa.

The metalloprotein SOD is an important antioxidant defense in almost all cells exposed to oxygen. SOD, together with glutathione (GSH), is one of the most important intracellular scavenger systems of spermatozoa and erythrocytes, able to reduce the superoxide anion (O_2_^−•^) to hydrogen peroxide (H_2_O_2_) [48]. CAT and GPX together with SOD represent the catalytic forces to protect cells, particularly the two cells considered, against excessive ROS production [49,50]. 

This review collects recent articles (2018–2023) focusing on ROS and major enzymatic biomarkers of OS in two mammalian cells: erythrocytes and spermatozoa. The aim is to provide an overview of the main methods used to measure them in small domestic ruminants.

## 2. Oxidative Stress in Erythrocytes 

Hematopoietic stem cells (HSCs) of bone marrow are multipotential cells that give rise to mature cellular elements of the blood (e.g., red blood cells (RBCs), neutrophils, monocytes, platelets, etc.) [51]. Erythropoiesis is the process of proliferation and progressive differentiation of HSCs into RBCs. Precursor cells undergo a series of division and differentiation steps until the nucleus is extruded (in mammals) [52]. Mature RBCs have no nuclei and organelles, and thus are not able to synthesize proteins. Therefore, the entire complement of functional proteins must be already present in mature erythrocytes. The RBCs membrane, essential for microcirculation and gas exchange [53], consists of a phospholipid bilayer with integral proteins associated. Between them, the glycoprotein membrane Band 3 protein (B3p) makes up to 50% of erythrocytes membrane proteins [41].

Several studies report that oxidative damage in erythrocytes leads to irreversible clustering of B3p [54,55]. In physiological conditions, this is an essential step of RBCs senescence and eryptosis processes, that result in cellular removal [42,43]. Such structural alteration, linked to oxidative damage, implies the removal of senescent erythrocytes [55]. Baralla et al. [18] reported that the sheep erythrocytes, damaged by the treatment with prooxidant molecules, cannot be eliminated in in vitro conditions, because they are not provided in in vivo blood circle, and then, they undergo hemolysis due to membrane damage. 

Oxidative damage has been demonstrated to decrease the survival and rheological properties of erythrocytes, affecting their homeostasis, which is tightly linked to B3p functions, particularly vulnerable to redox balance variations [55]. Alterations in erythrocyte deformability, caused by OS, underline the problems in the microcirculatory profile [56].

OS can damage membrane PUFAs, such as RBCs hemoglobin and several enzymes (particularly their sulfhydryl groups) [57]. Oxidation of hemoglobin alpha leads to the formation of hemichromes, which are proportional to hemolysis [58,59]. Interaction of oxidized hemoglobin with the membrane might lead to removal of RBCs through the increase in membrane rigidity, decrease in deformability and/or stimulation of membrane proteolysis [60]. These processes might be a cause of the increase in morphological changes and osmotic fragility of RBCs, making erythrocytes susceptible to phagocytosis [18,61].

After being release from bone marrow, RBCs circulate throughout the vascular system thousands of times before being removed, disassembled, and scavenged by macrophages. The RBC lifespan varies characteristically with species and the differences are at least partially explained by different metabolic rates. Comparative studies reveal that RBCs’ lifespan is positively correlated with species longevity and body mass. As a result, bigger mammals tend to live longer and have longer RBC lifespan than smaller species [60,62].

Several experimental studies reported that RBC lifespan in healthy animals may be influenced by RBCs’ antioxidant status. Kurata et al. [60,63] found that RBCs’ potential lifespan in several mammalian species is significantly correlated with the RBCs’ levels of SOD, GPX, and GSH.

Aged RBCs have shown compromised antioxidant capacity, including decreases in reduced glutathione, GSR activity and increased lipid peroxidation [60].

For all these reasons, the RBCs represent an ideal model to investigate oxidative damage in bio-membranes [64].

## 3. Oxidative Stress in Spermatozoa

The spermatozoon, a gamete generated in the male reproductive tract, moves to the female reproductive tract in order to fertilize an egg [65]. In mammals, sperm of the post testicular epididymal tract turns into spermatozoa capable of fertilization [50]. ROS have proven to be fundamental in sperm maturation and capacitation processes. However, until fertilization, oxidative damage is a threat to spermatozoa [50]. Membrane integrity, and in particular acrosome integrity, are essential for spermatozoa fertilizing capacity and for sperm penetration into the egg [66]. PUFA presence gives the membrane the fluidity and flexibility needed to engage in membrane fusion events associated with fertilization [27].

As a transcriptionally silent cell, spermatozoa largely depend on post-translational modifications of proteins to regulate their functions. Among these regulatory mechanisms, redox regulation plays a significant role, and the deregulation of redox homeostasis is a cause of male infertility [65].

Several technologies have been experimented to improve animal reproductive performances. Among them, the cryopreservation of semen, widely used in animals, is an energy-demanding activity, which generates ROS during the metabolic processes [67].

OS represents a major factor in sub lethal cryodamage of sperm in animal species, because, after semen dilution and cryopreservation, there is a significant reduction in the level of spermatozoa antioxidants, thus leading to enhanced susceptibility of these cells to peroxidative injuries [27,68].

In mammalian spermatozoa, ROS can attack important substrates, such as DNA, PUFA, and proteins [69]. Experimental studies revealed that OS breaks the DNA strand in spermatozoa, as detected via the comet assay [27,69].

The lipid peroxidation of PUFA in spermatozoa results in the loss of membrane functionality and integrity, and it varies across sperm regions [70,71]. For example, frozen/thawed ruminants spermatozoa showed more intense lipid peroxidation in the sperm flagellum with a greater PUFA content, with respect to the sperm head [71,72,73]. MDA is a cytotoxic aldehyde produced by lipid peroxidation and its assay provides important indications as an OS marker [27,74]. Researchers found that the use of antioxidants in the sperm freezing extender can reduce the destructive effects of ROS and can improve the post-thawing quality of goat spermatozoa [75].

Spermatozoa proteins are involved in key functions, such as sperm motility, capacitation, oocyte binding ability, and the acrosome reaction [76]. Significant redox-dependent modifications of proteins are thiol oxidation, tyrosine nitration, and S-glutathionylation. O’Flaherty et al. [77] reported that these reactions were associated with an impairment of sperm function leading to infertility, altering sperm motility in ruminants and equine, and sperm capacitation. Within proteins, the specific ratio of oxidated/reduced thiol groups (SS/SH) is essential to assure protein functions, but it can be affected by ROS [77]. Thus, the quantification of total thiols and/or GSH in cells is critical for monitoring the level of OS.

In cellular detoxification processes, the SOD-CAT-GPX catalytic triad represents the antioxidant defense against excessive ROS production in spermatozoa (Figure 2). GPX is the major representative of catalytic triad and its expression failure in the spermatozoa is correlated with infertility [50].

## 4. Assays for Measurement of ROS 

### 4.1. ROS Assay in Erythrocytes

Two recent papers for the measurement of ROS in ovine erythrocytes, using the most common ROS assays in RBCs (7 × 10^6^ RBCs/mL), with fluorescence detection, were reported [18,46]. In both assays, probe 2′,7′-dichlorodihydrofluorescein diacetate (H_2_DCF-DA) at the final concentration of 3 µM was used. Within the cell, the esterase cleaves the acetate groups of H_2_DCF-DA, resulting in the formation of the probe 2′,7′-dichlorodihydrofluorescein (H_2_DCF). Then, intracellular ROS oxidize the H_2_DCF, yielding the fluorescent product, 2′,7′-dichlorofluorescein (DCF). Fluorescence was measured using a fluorescent microplate reader at excitation and emission wavelengths of 485 and 535 nm, respectively. All fluorescence measurements (relative fluorescence unit—RFU) were corrected for background fluorescence [46] and expressed as RFU/10^6^ erythrocytes. In the two papers, the values of intracellular ROS for untreated RBCs of sarda sheep were ≈300 RFU/10^6^ erythrocytes. Given that H_2_DCF-DA is not cell-specific, some authors quantified fluorescence specifically on animal erythrocytes, removing leukocytes and platelets signal, using flow cytometry [78]. 

### 4.2. ROS Assay in Spermatozoa

Different spectrophotometric, fluorometric and chemiluminescent assay methods for intracellular ROS quantification in small domestic ruminant spermatozoa are summarized in Table 3.

Falchi et al. [79] used 2′,7′-dichlorodihydrofluorescein diacetate probe (H_2_DCF-DA, final concentration of 10 µM) for the assessment of intracellular ROS in spermatozoa of rams (Italy). Samples (600 × 10^6^ spermatozoa/mL) were analyzed via flow cytometric analysis. Fluorescence emission increased from a basal value of about 2 to a value of about 5 following sperm incubation at 4 °C for 96 h with or without increasing concentrations of cerium dioxide nanoparticles (CeO_2_ NPs). ROS concentrations ranged from 1.5 to 6 Rfu (relative fluorescence unit).

Merati et al. [80] measured, via a flow cytometer at 488 nm, the ROS level in spermatozoa of rams (4–5 years old, Iran). Briefly, 10 mL of 2′,7′-dichlorofluorescein diacetate (DCFH-DA, 25 mM), and 20 mL of dihydroethidium (DHE, 125 mM) were added to 1 × 10^6^ sperm suspension and incubated at 25 °C for 40 min for DCFH-DA and for 20 min for DHE. DCFH-DA is considered a general indicator of ROS, reacting with H_2_O_2_, ONOOֺ°, lipid hydroperoxides, while DHE is used extensively to monitor superoxide production. DHE upon reaction with superoxide anions forms a red fluorescent product (ethidium = HE) which intercalates with DNA. Green fluorescence (DCFH) was evaluated between 500 and 530 nm, and red fluorescence (HE) was assessed between 590 and 700 nm (excitation, 488 nm; emission, 525 and 625 nm). Data were expressed as the percentage of fluorescent spermatozoa. Propidium iodide (PI) has been used as a counter-stain dye for DCFH and Yo-Pro-1 as a counter-stain dye for HE. The results ranged from 22.8% to 56.1% for H_2_O_2_ and from 0.4% to 3.1% for O_2_^−^.

Reiten et al. [81] used spermatozoa (5 × 10^6^ spermatozoa/mL) of the Norwegian dairy goat breed (age 208–223 days, Hjermstad, Norway) for assaying ROS with flow cytometer. Hoechst 34580 (1.25 µM) and Mitotracker Orange CMTMRos (MO, 0.15 µM) (Invitrogen, ThermoFisher Scientific, Rodano, Italy) were used to eliminate non-spermatozoa events based on DNA and mitochondrial staining, respectively. PI (5 µg/mL) was used to discriminate between plasma membrane-intact and degenerated spermatozoa, while CellROX R Deep Red Reagent was used to assess levels of ROS as a measure of cellular oxidative stress. The results ranged from 10 to 60%. 

Escobar et al. [82] assayed ROS, with a spectrofluorimetric method using DCFH-DA, in rams spermatozoa (average age 10 months old, Brazil). The samples were incubated in the dark with 5 mL of DCFH-DA (1 mM). The oxidation of DCFH-DA to DCFH by the ROS was monitored. The fluorescence intensity emitted at 520 nm (488 nm excitation) was monitored 60 min after the addition of the probe. The results ranged from 10 to 100 Rfu. 

Gimeno-Martos et al. [83] assayed ROS in spermatozoa of Raza Aragonesa rams (2–4 years old, Zaragoza, Spain). Briefly, 5 × 10^6^ sperm/mL were stained with 5 µL of probes (H_2_DCF-DA, 20 µL and PI, 1.5 mM) after 15 min at 37 °C in the dark; the samples were fixed with 5 µL formaldehyde (0.5% in water) and analyzed via flow cytometry (filter 525 and 675 nm). Intracellular ROS levels ranged between 3.9 and 11.2 Rfu.

Liu et al. [84] assayed ROS in spermatozoa of Guanzhong dairy goats (2–3 years old, Shaanxi, China) using DCFH-DA. Briefly, 1 × 10^6^ spermatozoa were washed in PBS and were incubated with 100 mM DCFH-DA at 37 °C for 30 min. The fluorescence of DCF was measured on a flow cytometer at a wavelength of 485/535 nm. The results ranged from 8.36 to 53.83% for DCF-positive cells.

Lv et al. [85] used goats (2–3 years old, Kunming, China) for the collection of semen. Briefly, 1 × 10^6^ spermatozoa/mL was mixed with H_2_DCF-DA (20 μM) and 50 μL PI (50 μg/mL). Then, the samples were incubated for 60 min at room temperature in darkness. The spermatozoa suspension was analyzed using flow cytometry. Results are expressed as percentage of viable spermatozoa with less ROS production (identified by H_2_DCF-DA negative and no PI staining) and were ranged from 54 to 67%. 

Zarepourfard et al. [86] used male goats (5–6 years old, Isfahan, Iran) to carry out semen collection employing flow cytometer to determine the ROS content. The percentage of ROS positive spermatozoa was measured following incubation of 1 × 10^6^ sperm/mL with 5 mM of DCFH-DA for 30 min at room temperature. Live sperm produce a detectable amount of physiological ROS and therefore, the sperm may become DCF-positive. ROS results ranged from 38.7% to 62.0%.

Kumar et al. [87] measured oxidative status in spermatozoa of sexually mature and healthy goats (2–3 years old). The sperm (10 × 10^8^ spermatozoa/mg protein) was mechanically lysed by liquid nitrogen, centrifuged and the supernatant was used for the analysis. OS was assessed by monitoring the concentration of the *N*,*N*-diethylparaphenylendiamine (DEPPD) radical cation. The formation of this radical cation is not only due to hydroperoxides present in the sample, but also other oxidizing agents. ROS results ranged between 0.16 ± 0.011 and 0.74 ± 0.028 mg H_2_O_2_/10^8^ spermatozoa. [87].

Anzalone et al. [88] used a dichloromethyl derivate of H_2_DCFDA (5-(and 6-)chloromethyl-2′,7′-dichlorodihydrofuorescein diacetate (CM-H_2_DCFDA, final concentration of 3 µM) as a probe for ROS determination in intact ram spermatozoa. CM-H_2_DCFDA is sensitive to a wide range of ROS (H_2_O_2_, ONOO-, superoxide anion, and hydroxyl). Fluorescence measurements (excitation and emission wavelengths were 490 nm and 520 nm, respectively) were carried out in a multimode plate reader during overnight incubation (38 °C). The kinetic of the probe oxidation rate was evaluated from the slope of the fluorescence emission intensity in the initial 10 min of the total recording time. The results are expressed as mean and standard deviation of fluorescence intensity slope as a function of time (DF/min) in the linear region (corresponding to the first 10 min of the kinetic measurement) and the authors found ROS equal to ≈ 63 DF/min when spermatozoa were diluted in PBS.

The same CM-H_2_DCFDA fluorescent probe for ROS quantification in ovine spermatozoa was also performed using a cytofluorometer detector [89,90]. Del Olmo et al. [89] found that the 3.3% ÷ 4.0% ± 0.2% of live spermatozoa positive to probe in ram spermatozoa (10^8^ cells/mL) using flow cytometer. Vašíček et al. [90] measured ROS levels in ram semen samples (1 × 10^6^ spermatozoa in PBS) via flow cytofluorimetry using four different fluorescent probes: a green dye nonspecifically indicating the presence of intracellular ROS (CM- H_2_DCFDA, final concentration of 500 nM), a red superoxide indicator (DHE, final concentration of 2 µM), a red mitochondrial superoxide indicator (MitoSOX™, final concentration of 500 nM), and a green lipid peroxidation sensor (BODIPY™, final concentration of 200 nM) [90]. They reported 25 ÷ 40% positive cells to CM- CM-H_2_DCFDA, 15 ÷ 25% positive cells to DHE, 10 ÷ 20% positive cells to MitoSOX and 8 ÷ 20% positive cells to BODIPY. The authors reported that the unspecific CM-H_2_DCFDA probe had higher proportion of ROS positive spermatozoa than the specific probes. Thus, CM-H_2_DCFDA seems to be a suitable probe for rapid unspecific detection of ROS production in ram semen samples prior to their further processing [90].

Zarei et al. [91] evaluated the intracellular H_2_O_2_ concentration as an index of ROS, using DCFH-DA as a fluorescent probe (final concentration of 25 µM). DCFH-DA probe was added to ram sperm (Zandi sheeps 3 ÷ 5 × 10^6^ spermatozoa/mL 200 cells/slide) and the fluorescence intensity was evaluated using a fluorescent microscope (490 nm/520 nm, excitation/emission wavelengths, respectively). ROS concentration was expressed as the percentage of probe-positive cells out of the total spermatozoa counted. The authors [91] found 10% of ROS in fresh semen and 24 ÷ 30% in frozen–thawed semen. 

O’Brien et al. [92] detected ROS in sperm (25 × 10^6^ sperm/mL 200 sperm cells/slide) of four different ovine species (ram (*Ovis aries*), mouflon (*Ovis musimon*), buck (*Capra hircus*), and Iberian ibex (*Capra pyrenaica*) from Madrid (40°25′ N), Spain) using a CellROX R green probe (final concentration of 5 µM) by an epifluorescence microscope with a triple band-pass filter (450 nm/490 nm, excitation/emission, respectively). This fluorescent probe penetrates the cell and when oxidized by intracellular free radicals, binds to DNA, emitting a more intense green fluorescence. The response to the stress of each species was illustrated by calculating a stress resistance ratio (SR) for the sperm variables: SR = (value after stress/value before stress) × 100 [92]. The ROS in sperm were evaluated before the stress condition, after refrigeration at 15 °C for 20 h, and subsequent incubation at 38.5 °C for 2 h. ROS increases significantly in three of the four sheep species after incubation at 15 °C and also after subsequent incubation at 38 °C, increasing from SR basal values of 100 to SR maximum values of 350.

Jiménez et al. [93] detected intracellular superoxide (O_2_^−•^) levels in ram semen (5 × 10^6^ cells/mL) from Raza Aragonesa rams (2–4 years old) located in Zaragoza, Spain, using DHE (final concentration of 4 µM) and Yo-Pro-1 (final concentration of 40 nM) fluorochromes. DHE is permeable to cells, and it is oxidized by O_2_^−•^ to the red fluorescent compound ethidium (E) and detected using flow cytometer. Yo-Pro-1 labels non-viable cells with green fluorescence. A percentage of 10% of viable spermatozoa with high superoxide levels (Yo-Pro-1^−^/E^+^) was found, although the values increased (35%) with in vitro addition of calcium ionophore, and then returned to lower values (20%), with the concomitant presence of melatonin and its antioxidant action.

Wang et al. [94] used a commercial kit (Beyotime Institute of Biotechnology, Shanghai, China) to determine ROS content in ram semen (2.0 × 10^9^/mL), using DCFH-DC as a probe. The fluorescence intensity was monitored using a multifunctional microplate reader (PerkinElmer Inc., NYSE: PKI, Waltham, MA, USA) at Ex/Em = 488/525 nm). The results found ranged from 2500 to 3500 rfu.

Yu Gao et al. [95] studied old Suffolk White rams (age of 1–2 years Beijing, China) and the in vitro effect of leptin on their sperm quality. Intracellular ROS of sperm was assessed with a ROS Assay Kit (SAS, Beijing, China) containing H_2_DCFDA according to the manufacturer’s instruction. Briefly, 1 µL of H_2_DCFDA and 5 µL of PI were added into the sperm suspension and mixed well, following incubation at 37 °C for 30 min in the dark. The samples were centrifuged and suspended with PBS. Then, the samples were detected using a flow cytometer. ROS levels determined were 0.25 ÷ 0.8%.

Shahat et al. [40] studied the effects of melatonin or L-arginine on the quality of frozen-thawed sperm from heat-stressed Dorset rams (3–4 years old, Canada). For total ROS, ~3 × 10^6^ sperm were centrifuged at 400× *g* for 5 min, yielding a working concentration of 1 × 10^5^ sperm/sample. Then, total ROS was analyzed using a total ROS Assay Kit 520 nm (Invitrogen Life Technologies, Carlsbad, CA, USA) on BD™ LSR II flow cytometer (BD Biosciences, Oxford, UK) at 520 nm using the FITC channel. The flow rate was set to 200 events/s, with 10,000 events counted per sample and all samples tested in duplicate. The ROS detection solution contains a fluorescent probe that stains ROS in sperm, with results expressed as a percentage of sperm that had high ROS activity. The values found ranged from 62.1 ± 1.0 to 79.4 ± 1.4 ROS%.

Liang et al. [96] studied the effects of four extenders for semen cryopreservation (Andromed^®^ (Andr^®^), Optidyl^®^ (Opt^®^), P3644 Sigma l-phosphatidylcholine (l-α SL), and skim milk-based (milk) extenders) on the post-thaw quality and fertility of goat semen (Yunshang black bucks, weight 80–100 kg, aged between 2 and 3 years). The intracellular concentrations of ROS in frozen–thawed sperm were determined by staining with 2′, 7′-dichlorodihydrofluorescein diacetates (H_2_DCFDA) with flow cytometry analysis. Briefly, 500 μL of semen sample was diluted in the TALP buffer to obtain a final concentration of 1 × 10^6^ sperm/mL and mixed with 0.5 μL H_2_DCFDA (final concentration of 20 μM) and 50 μL PI (50 μg/mL). Then, the samples were incubated for 60 min at room temperature in the darkness. The sperm suspension was analyzed using flow cytometry. The percentage of viable sperm with low ROS was identified by H_2_DCFDA negative and no PI staining. The green fluorescence from FITC-PSA, Annexin-V-FITC, and H_2_DCFDA was detected on a FacStar-plus flow cytometer (FAC-SCalibur) on the FL1 photodetector (530/30 BP filter). The values found ranged from 45 to 60 ROS%.

Longobardi et al. [97] studied the effect of crocin before cryopreservation in the extender for bucks/breed (2–4 years age, Italy) semen. ROS levels were measured as fluorescence emission of DHE. This probe is a cell-permeable compound oxidized by superoxide anion (O_2_^−•^) to ethidium bromide that binds to DNA and emits red fluorescence. DHE (2 µM) was added to frozen–thawed sperm samples and incubated in the dark, at room temperature, for 20 min. Emission was monitored at 570 nm, using a plate reader (GloMax^®^-Multi Detection System-Promega, Milano, Italy). ROS levels were evaluated as arbitrary units of fluorescent signal. ROS values ranged between 437.8 ± 4.9 and 347.6 ± 2.8 fluorescence intensity. 

Monteiro et al. [98] evaluated the effect of antifreeze protein type III (AFP III) on the freezing of epididymal spermatozoa of goat (Brejo da Madre de Deus-PE, Brazil). For intracellular ROS levels analysis, performed with the Amnis ImageStream Mark II flow cytometer (EMD Millipore Corp., Lausanne, Switzerland), 5 μL of CM-H_2_DCFDA (50 μM in PBS) were added to the samples at 37 °C for 30 min. These were subsequently diluted with 1 mL of PBS and centrifuged (100× *g*/5 min) to remove unbound fluorochrome, and the sediment was resuspended with 40 μL of PBS. Then, 5 μL of PI was added to the sample, incubated at room temperature (5 min) and analyzed. The results were expressed as a percentage of viable cells with high levels of ROS (DCFDA+/PI). Values ranged from 79.02 ± 35.51 to 88.67 ± 23.13 ROS %. 

Esmaeilkhanian et al. [99] assessed the efficacy of Mito-TEMPO on post-thawed goat sperm quality using Saanen goats in breeding season for sperm collection. Dichlorofluorescein diacetate (DCFH-DA) evaluated the intracellular H_2_O_2_ concentrations as an index of ROS. Semen samples were rinsed with PBS to a concentration of 3–5 × 10^6^ spermatozoa/mL. DCFH-DA (25 μM) was added to the sperm suspension and incubated for 40 min at 25 °C. After washing, 2 μL of PI were added to the semen, and samples were analyzed via flow cytometry. Obtained values were ranged from 20 to 25 ROS %.

Nazari et al. [100] studied the effect of trehalose and pentoxifylline in diluents on cooled and frozen–thawed Markhoz goat sperm (3–4 years old of age and weighing 60–65 kg, Sanandaj, Iran). ROS flow cytometry (excitation: 488 nm; emission: 525–625 nm) was carried out using two specific dyes, DCFH-DA and DHE, to detect the intracellular H_2_O_2_ and O_2_^−•^ in 1 × 10^6^ spermatozoa. Data were expressed as the percentage of fluorescent sperm and PI was used as a counterstain dye for DCFH. ROS % values ranged from 19.24 to 52.78 and from 1.73 to 2.61 for H_2_O_2_ and O_2_^−•^, respectively.

Lv et al. [101] studied the effects of antifreeze protein (AFP) on frozen spermatozoa of black goats (2–3 years old Kunming, Yunnan province, China). The ROS ratios were measured with a flow cytometer by incubating spermatozoa with H_2_DCFDA. In brief, a 500 µL semen sample was first mixed with the TALP buffer to a final concentration of 1 × 10^6^ spermatozoa/mL. Then, H_2_DCFDA (final concentration of 20 mM) and PI (50 mg/mL) were added to the above solution. Finally, mixed samples were incubated in darkness for 60 min at room temperature. Flow cytometry was used to examine the spermatozoa suspension. The percentage of viable spermatozoa with low ROS was identified by negative H_2_DCFDA and no PI staining. Values ranged from 60 to 65 ROS %.

Sun et al. [102] measured the levels of ROS in goat semen from Chongming White goats (3–5 years old) located in Shanghai, China, during spring to early summer, by a chemiluminescence assay using luminol (5-amino-2,3-dihydro-1,4-phthalazinedione). ROS levels were higher in the post-thawed spermatozoa containing soybean lecithin (SL) added to the extender at different concentrations, than in the control (egg yolk (EY) present in the extender). Only SL at 2% brought the ROS values equal to the control (5.5 IU/mL = intensity light units/mL).

Mehdipour et al. [103] investigated the effects of rosiglitazone on rams semen after cryopreservation on the quality of thawed sperm. Sperm from sexually mature Ghezel rams (3 to 4 years of age, Tabriz, Iran.) were sampled. After washing spermatozoa with PBS and their centrifugation at 300× *g* for 7 min, the supernatant was removed and luminol was added to measure ROS production. The results were expressed as 10^3^ counted photons per minute (cpm) per 10^6^ spermatozoa and ranged from 2 to 4.5 × 10^3^ cpm/10^6^ spermatozoa.

**Table 3 animals-13-02300-t003:** Different methods for ROS assay in spermatozoa of small domestic ruminants.

Method	Instrument	Probe	Specie	Detected ROS	Concentration ROS	Refs.
ABSORBANCE	Spectrophotometer	DEPPD	Goat	Hydroperoxides Other oxidizing agents	0.16 ÷ 0.74 mg H_2_O_2_/10^8^ spermatozoa	[87]
FLUORESCENCE	Multimode plate reader	CM-H_2_DCFDA	Ram	Hydrogen peroxide Superoxide radical hydroxyl radical	≈30 ÷ 65 rfu/min	[88]
		Commercial kit (CA1410) by Solarbio	Goat	N.S.	397.6 ÷ 461.8 U/mL	[104]
		DCFH-DC	Bucks	Hydrogen peroxide	3000 ÷ 3800 rfu	[105]
		DHE	Bucks Breed	Superoxide radical	437.8 ÷ 347.6 rfu	[97]
	Fluorescent microscope	DCFH-DA DCFH-DC	Ram	Hydrogen peroxide	10 ÷ 30%	[91,94]
	Epifluorescence microscope	CellROX R green	Ram Mouflon Buck Iberian ibex	Intracell.FreeRadicals	100 ÷ 350 SR (stress resistance ratio)	[92]
	Flow Cytometer	H_2_DCF-DA	Ram	Hydrogen peroxide	1.5 ÷ 6 rfu	[79]
		DCFH-DA/PI DHE/Yo-Pro-1	Ram	Hydrogen peroxide Superoxide radical	22.8 ÷ 56.1% 0.4 ÷ 3.1%	[80]
		CellROX Rgreen/PI	Goat	Intracell.FreeRadicals	10 ÷ 60%	[81]
		DCFH-DA	Ram	Hydrogen peroxide	10 ÷ 100 rfu	[82]
		H_2_DCF-DA/PI	Ram	Hydrogen peroxide	3.9 ÷ 11.2 rfu	[83]
		DCFH-DA	Goat	Hydrogen peroxide	8.4 ÷ 53.8%	[84]
		H_2_DCF-DA	Goat	Hydrogen peroxide	54 ÷ 67%	[85]
		DCFH-DA	Goat	Hydrogen peroxide	38.7 ÷ 62.0%	[86]
		CM-H_2_DCFDA	Ram	N.S	3.3 ÷ 4.0%	[89]
		CM-H2DCFDA	Ram	N.S.	25 ÷ 40%	[90]
		DHE		Superoxide radical	15 ÷ 25%	
		MitoSOX™		Mitochondrial superoxide	10 ÷ 20%	
		BODIPY™		Lipid peroxides	8 ÷ 20%	
		DHE	Ram	Superoxide radical	10 ÷ 45%	[93]
		H_2_DCF-DA Assay Kit (SAS, China)	Ram	Hydrogen peroxide	0.25 ÷ 0.8%	[95]
		DCFH-DA/PI	Goat	Hydrogen peroxide	20 ÷ 25%	[99]
		Commercial kit (Invitrogen Life Technologies)	Ram	N.S.	62.1 ÷ 79.4%	[40]
		CM-H_2_DCFDA/PI	Goat	Hydrogen peroxide	79.0 ÷ 88.7%	[98]
		DCFH-DA DHE/PI	Goat	Hydrogen peroxide Superoxide radical	19.2 ÷ 52.8% 1.7 ÷ 2.6%	[100]
		H_2_DCFDA/PI	Goat	Hydrogen peroxide	60 ÷ 65%	[101]
		H_2_DCF-DA	Goat buck	Hydrogen peroxide	45 ÷ 60%.	[96]
CHEMOLUMINESCENCE	Chemoluminescence reader	Luminol	Goat	N.S.	34.9 ÷ 60.6%	[102]
		Luminol	Ram	N.S.	2 ÷ 4.5 × 10^3^ cpm/10^6^ sperma	[103]

DEPPD = *N*,*N*-diethylparaphenylendiamine radical cation; CM-H_2_DCFDA = (5-(and 6-)chloromethyl-2′,7′-dichlorodihydrofuorescein diacetate; DCFH-DA = 2′-7′dichlorohydrofluorescin diacetate; H_2_DCFDA = 2’,7’-dichlorodihydrofluorescein diacetate; DCF-DA = 2′-7′dichlorofluorescein diacetate; DHE = dihydroethidium (hydroethidine); MitoSOX = MitoSOX™ Red mitochondrial superoxide indicator; BODIPY = BODIPY™ 581/591 C11, a green lipid peroxidation sensor; Luminol = 5-amino-2,3-dihydro-1,4-phthalazinedione; PI = Propidium Iodide. N.S. = Not specified intracellular ROS. Rfu = Relative fluorescence unit.

As can be seen in Table 3, in recent years, most researchers employed fluorescent methods using mostly commercial kits, for ROS quantification. Only two papers with luminescent methods and one with spectrophotometric method are reported.

## 5. Assays for the Measurement of Enzymatic Antioxidants 

### 5.1. SOD 

#### 5.1.1. SOD in Erythrocytes 

Different methods for the measurement of SOD in erythrocytes of small domestic ruminants are summarized in Table 4. 

In 2021, Pasciu et al. [46] investigated oxidative response under in vitro conditions using high doses of glycerol on the RBC of Sardinian sheep (Sassari, Italy). Blood cells were treated with HBSS containing 0.1% Triton X100. SOD activity in cellular extracts was enzymatically detected at 470 nm using the xanthine and xanthine oxidase methods. The SOD levels ranged from 50 to 80 U/106 erythrocytes.

Mousaie et al. [106] measured SOD activity in erythrocytes of male lambs (Jiroft, Iran), 28.5  ±  2.6 kg of body weight. RBC hemolysate was prepared according to manufacturer’s protocol (Randox Ransod kit, Randox Laboratories Ltd., Antrim, UK). The mean SOD level in erythrocytes was 1307.2 ± 29.8 unit per gram Hb (U/g Hb).

Tao et al. [107] evaluated the effect of copper sulphate poisoning on the morphological and functional characteristics of goat RBCs: for this in vitro study, five goats which were 10–14 months old (Xinjiang Uygur Autonomous Region China) were used. Erythrocytes were hemolyzed under hypotonic conditions. SOD activity was measured in terms of inhibition of the decrease in nitro-blue tetrazolium (NBT) by superoxide. One unit of SOD activity was determined as the amount of enzyme that caused a reduction in half the maximum suppression of NBT; this reduction was acquired by measuring the absorbance at 550 nm. The SOD obtained was 200 ÷ 300 U/mg protein.

In 2022, Zheng et al. [49] measured the antioxidant effect of flavonoids extracted from Chinese herb mulberry leaves in an in vitro study on sheep erythrocytes for analyzing the activity of antioxidative enzymes. Commercial kits (Nanjing Jiancheng Institute of Biotechnology of China) were used to assay SOD, GPX and CAT. Blood cells were collected in tubes with heparin anticoagulant and washed three times with PBS (pH 7.4), lysed by adding approximately 4 times the volume of ultrapure water and centrifuged (1200× *g*, 10 min, 4 °C) after 10 min of an ice bath. The obtained cell lysate was stored at −80 °C until analysis. Reported values, were of 2 ÷ 13 U/mg protein, 1 ÷ 5.5 U/mg protein and 10 ÷ 65 U/mg protein for SOD, CAT and GPX, respectively.

Sousa et al. [108] used a commercial Randox kit (RANSOD) to measure SOD levels in RBCs of non-pregnant Santa Inês crossbred sheep (weighing on an average 52.89 ± 6.7 kg, semi-arid, UFERSA, Brazil). Blood samples were centrifuged and subsequently, RBCs were washed three times with phosphate solution (PBS 10%). Then, a freeze–thaw cycle was performed at −20 °C for 10 min, followed by melting at room temperature for another 10 min, repeating the process for two consecutive times. The obtained hemolysate was stored at −40 °C until analysis. The mean found SOD value was 11.9 ± 4.12 U/g Hg.

Seifalinasab et al. [109] studied the effect of dietary treatments with chromium supplementation on male lambs (Jiroft, Iran) (8–9 months old, average body weight of 31.9 ± 1.2 kg). Blood sample was taken in heparinized tubes and SOD activity was measured with a commercial kit Ransod (Randox Laboratories Ltd., Antrim, UK). The method employs xanthine and xanthine oxidase to generate superoxide radicals which react with 2-(4-iodophenyl)-3-(4-nitrophenol)-5-phenyltetrazolium chloride to form a red dye (470 nm). The SOD activity is then measured by the degree of inhibition of this reaction. The mean value for SOD was 1127.37 ± 48.75 U/g Hb.

**Table 4 animals-13-02300-t004:** Spectrophotometric methods for SOD detection in erythrocytes of small domestic ruminants.

Specie	Method	Erythrocytes Lysis	SOD Concentration	Refs.
Sheep	Commercial Kit (Nanjing Jiancheng Institute of Biotech., Nanjing, China)	Ultrapure water	2 ÷ 13 U/mg protein	[49]
	Colorimetric assay: measure at 470 nm using xanthine and xanthine oxidase	HBSS with Triton X100	50 ÷ 80 U/106 erythrocyte	[110]
	Commercial kit Ransod (Randox Lab. Crumlin, Co., Antrim, UK) using xanthine and xanthine oxidase	Freeze–thaw	11.9 U/g Hb	[108]
Ram lamb	Commercial kit Ransod, (Randox Lab., UK) using xanthine and xanthine oxidase	Manufacturer protocols	1307.2 U/g Hb	[106]
	Commercial kit Ransod (Randox Lab., UK) using xanthine and xanthine oxidase	N.S.	1127.4 U/g Hb	[109]
Goat	Colorimetric assay: using NBT	Hypotonic conditions.	200 ÷ 300 U/mg protein	[107]

N.S. = not specified. NBT = nitro-blue tetrazolium. HBSS = Hanks′ balanced salt solution.

As can be seen in Table 4, all analyzed studies reported spectrophotometric methods, and many of them used commercial kits.

#### 5.1.2. SOD in Spermatozoa

In Table 5, a summary of SOD assays in spermatozoa of small domestic ruminants is reported. In particular, Hashem et al. [111] determined the total SOD activity in spermatozoa of Qezel rams (3–4 years of age, Urmia, Iran), assaying the auto oxidation and illumination of pyrogallol at 420 nm for 3.5 min. One-unit total SOD activity was calculated as the amount of protein causing 50% inhibition of pyrogallol autooxidation. A blank without treated sample was used as a control for non-enzymatic oxidation of pyrogallol in Tris–EDTA buffer (50 mM Tris, 10 mM EDTA, pH 8.2). The total SOD activity was expressed as units per milligram of protein (unit/mg protein) in the spermatozoa samples and the authors found 50 ÷ 60 Unit/mg protein for SOD spermatozoa of ram. 

Ren et al. [112] measured SOD, CAT, and GSH-Px activities from spermatozoa of Cashmere goats (aged 3–5 years old, Yulin, Shaanxi, China). The samples from the thawed frozen semen were centrifuged at 1600× *g* for 5 min at 25 °C, and the supernatant was discarded. The sperm pellet was re-suspended with Triton X-100 (1%) for 20 min for extracting enzymes, was centrifuged at 4000× *g* for 30 min at 25 °C and its supernatant was collected for follow-up determination. The antioxidant enzyme activities of SOD, CAT, and GPX were detected using a commercial enzyme detection kit (Nanjing Jiancheng Bioengineering Institute, Nanjing, China). The results were 200 ÷ 250 U/mL for SOD, 1.8 ÷ 3.10 U/mL for CAT, and 18 ÷ 33 U/mL for GPX.

Liu et al. [113] evaluated the antioxidant effects of a diet with different zinc levels on spermatozoa of Liaoning Cashmere goats (body weight of 56.2 ± 2.45 kg and age of 3 years, Shenyang, China). Cells were separated from semen fluid and washed by resuspending in PBS. After three centrifugations, 1 mL of Triton X-100 (0.1%) was added to spermatozoa, and the samples were centrifuged again at 8000 rpm for 30 min in a refrigerated centrifuge. The supernatant was analyzed using the Bradford method, and SOD activity was determined using a commercial assay kit (Nanjing Jiancheng Bioengineering Institute, China). The SOD values ranged from 18 to 30 U/mg protein.

Al-Mutary et al. [114] studied the effect of resveratrol on antioxidant enzyme activities of ram sperm during cooling storage. Ram spermatozoa (50 × 10^6^/mL) were grinded using satirized Sigma rods (pestles for 2 mL tubes), and then centrifugated and analyzed for SOD evaluation using a colorimetric method that measured its ability to inhibit the phenazine methosulphate reduction in nitroblue tetrazolium dye. SOD levels, measured at 560 nm, ranged from 112.75 ± 5.80 U/mL to 180.09 ± 12.55 U/mL.

Mortazavi et al. [115] evaluated SOD activity by determining the extent of inhibition of pyrogallol autoxidation. Ram spermatozoa (25 × 10^6^/mL) was electronically homogenized at 5.000 ÷ 6.000 g for 5 min. The assay data were normalized considering the concentration of spermatozoa. The authors found 24.07 ± 1.48 U/mL ÷ 31.45 ± 5.13 U/mL of SOD.

Žaja et al. [116] investigated the effect of melatonin on antioxidative enzyme activity in the seminal plasma and spermatozoa of French Alpine bucks (with body weight ranging from 40 to 60 kg, aged 1.5 to 4 years, Varazdin, Croatia) during the nonbreeding season. The samples of spermatozoa were prepared as cellular lysates. The frozen samples were thawed and resuspended in cooled distilled water, then stored in a refrigerator at 4 °C for 10 min. The samples were then centrifuged at 2400× *g* for 5 min, and the obtained supernatants were used for analyzing the activity of antioxidative enzymes. SOD activity was determined using a commercial kit (Randos, Randox Laboratories) that uses xanthine and xanthine oxidase for generating superoxide radicals which react with 2-(4-iodophenyl)-3-(4-nitrophenol)-5-phenyltetrazolium chloride to form a red formazan dye. SOD activity was spectrophotometrically measured at 505 nm observing the degree of inhibition of this reaction. The activity of SOD in spermatozoa was 100 ÷ 600 U/g of protein. 

Saberivand et al. [117] assayed SOD activity in Romanov ram spermatozoa (Tabriz, Iran), based on the inhibition of nitro blue tetrazolium (NBT) reduction by superoxide radicals to blue-colored formazan. The reaction was followed by reading at 560 nm. Semen hemolysate was prepared by treating cells with ethanol and chloroform, and, after centrifugation, the supernatant was used for the assay. SOD activity is defined as the amount of enzyme required to inhibit the reduction in NBT by 50% detectable via the spectrophotometer at 560 nm and results were reported as U/mg protein (76.20 ÷ 118.60 U/mg prot).

Wang et al. [94] used a WST-8 kit (Beyotime Institute of Biotechnology, Shanghai, China) for measuring SOD activity in ram. Samples of semen (2.0 × 10^9^/mL) were centrifuged at 12,000× *g* for 5 min, and after incubation with WST-8/enzyme working solution, the absorbance at 450 nm was determined. SOD activity was 110 ÷ 220 U/mg protein. One unit of total SOD activity was calculated as the amount of protein causing 50% inhibition of pyrogallol autooxidation.

Also, Shayestehyekta et al. [118] studied the effect of melatonin on fresh epididymal spermatozoa of ram (2–5 years old, Kermanshah, Iran) after post-mortem recovery under normal and oxidative stress conditions under liquid preservation (4 °C) at different times (24, 48 and 72 h). To assess SOD activity, a SOD assay kit (TPR INNOVATIVE) was used after ultrasonic lysis on the ice of spermatozoa pellets and centrifugation at 6000 rpm for 10 min. The activity of SOD was 30 ÷ 70 U/mL.

Zhang et al. [105] studied the effects of proline supplementation on goat sperm (healthy Laoshan bucks aged between 1.5 and 2 years, Qingdao, China). SOD activity was assayed with a microplate reader at 450 nm, using a commercial kit (A001-3-2, Nanjing Jiancheng Bioengineering Institute) after ultrasonication on ice. The values found were 2.5–3.5 U/µg protein.

Zou et al. [104] explored the effect of bovine seminal plasma replacing different doses of dairy goat seminal plasma on semen freezing quality of Guanzhong dairy goat (2.5–3 years old, Lantian, Shaanxi Province, China). Thawed semen was used for SOD, and ROS assay using commercial kits (BC 0175, and CA1410) from Solarbio and the amount was found to be 165.99 ÷ 199.78 U/mL for SOD (Table 5), and 397.58 ÷ 461.79 U/mL for ROS (Table 3).

Li et al. [119] investigated the cryoprotective melatonin effect on the semen of healthy adult Huang-Huai rams (2–4 years of age) (Dingyuan County, Anhui province, China). Semen was diluted using two types of cryopreserved media (with and without melatonin) at a final sperm concentration of about 3 × 10^8^/mL. Fresh and cryopreserved semen pellet was sonicated on ice using a high intensity ultrasonic processor (Scientz) in pre-cooled sodium chloride injection (0.9% NaCl) (*m*/*v*) and 1% Triton X-100, followed by centrifugation at 5000× *g* for 10 min at 4 °C. The total SOD concentration, determined using a commercial assay kit (A001–1-2; Nanjing Jiancheng, China), was about 45 ÷ 60 U/mg protein.

**Table 5 animals-13-02300-t005:** Spectrophotometric methods for SOD detection in spermatozoa of small domestic ruminants.

Specie	Method	Erythrocytes Lysis	SOD Concentration	Refs.
Sheep	Commercial Kit (Nanjing Jiancheng Institute of Biotech., China)	Ultrapure water	2 ÷ 13 U/mg protein	[94]
	Colorimetric assay: measure at 470 nm using xanthine and xanthine oxidase	HBSS with Triton X100	50 ÷ 80 U/10^6^ erythrocyte	[97]
	Commercial kit Ransod (Randox Lab. UK) using xanthine and xanthine oxidase	Freeze–thaw	11.9 U/g Hb	[95]
Ram lamb	Colorimetric assay: measure at 470 nm using pyrogallol	N.S.	50 ÷ 60 U/mg protein	[111]
	Commercial kit Ransod, (Randox Lab., UK) using xanthine and xanthine oxidase	Manufacturer protocols	1307.2 U/g Hb	[92]
	Commercial kit Ransod (Randox Lab., UK) using xanthine and xanthine oxidase	N.S.	1127.4 U/g Hb	[96]
Goat	Commercial Kit (Nanjing Jiancheng Institute of Biotech., China)	Triton X-100 (1%)	200 ÷ 250 U/mL	[112]
	Colorimetric assay: using NBT	Hypotonic conditions.	200 ÷ 300 U/mg protein	[93]

NBT = nitro-blue tetrazolium. N.S. = not specified.

As can be seen in Table 5, all analyzed studies reported spectrophotometric methods, with many of them using commercial kits.

### 5.2. CAT, GPX, and GSR

#### 5.2.1. CAT, GPX, and GSR in Erythrocytes 

In the last five years, as reported in Table 6, only Zheng et al. [49] measured the antioxidant activity of enzymes GPX and CAT within erythrocytes. They used a commercial kit (Nanjing Jiancheng Institute of Biotechnology of China) to in vitro assay the antioxidant effect of Chinese herb mulberry leaves. Blood cells, collected in tubes with heparin anticoagulant, were washed three times with PBS (pH 7.4), lysed by adding ultrapure water and centrifuged (1200× *g*, 10 min, 4 °C) after 10 min of an ice bath. The levels of CAT and GPX ranged from 1 ÷ 5.5 U/mg protein to 10 ÷ 65 U/mg protein, respectively.

#### 5.2.2. CAT, GPX, and GSR in Spermatozoa 

In Table 6, a summary of CAT, GPX, and GSR assays in spermatozoa of small domestic ruminants is reported. In particular, Žaja et al. [116] investigated the effect of melatonin on the antioxidative activity of enzymes GPX, GSR, and CAT in French Alpine bucks spermatozoa (body weight ranging from 40 to 60 kg, aged 1.5 to 4 years Varazdin, Croatia). The cellular lysates were centrifuged at 2400× *g* for 5 min, and the obtained supernatants were used for analyzing the antioxidant activity of GSR, GPX, and CAT, using commercial kits (glutathione reductase kit, Randox Laboratories; Ransel kit, Randox Laboratories, Crumlin, UK; and Cayman Catalase Assay kit, Cayman Chemical Company, Ann Arbor, MI, USA, respectively). The absorbance was measured spectrophotometrically at 340 nm for both GSR and GPX and at 540 nm for CAT, using a Microtiter plate reader (Human, Wiesbaden, Germany). The obtained results were 2000 ÷ 3500 U/g protein for GSR, 3200 ÷ 4000 U/g protein for GPX and 300 ÷ 750 nmol/min/g protein for CAT.

Liu et al. [113] evaluated the GPX and CAT levels in spermatozoa, following the procedure already reported in the paragraph of SOD. The activities of GPX and CAT were determined using commercial assay kits (Nanjing Jiancheng Bioengineering Institute, Nanjing, China). The obtained results were 20 ÷ 80 U/mg protein for GPX and 2.5 ÷ 3 U/mg protein for CAT.

Li et al. [119] investigated the antioxidative enzyme activities in spermatozoa of Huang-Huai ram (2–4 years of age, Dingyuan county, Anhui province, China), followed by in freezing medium supplemented with melatonin. Sperm pellet (3 × 10^8^/mL) was sonicated on ice using a high intensity ultrasonic processor (Scientz) in pre-cooled sodium chloride injection (0.9% NaCl) (*m*/*v*) and 1% Triton X-100, followed by centrifugation at 5000× *g* for 10 min at 4 °C. GPX and CAT activities were measured using commercial kits (A005–1-1 and A007–1-1, Nanjing Jiancheng, China, respectively) and the absorbance was read at 412 nm (GPX) and 405 nm (CAT). GPX and CAT levels were 10 ÷ 16 U/mg protein, and 0.12 ÷ 0.20 U/mg protein, respectively.

Zou et al. [104] explored the GPX levels in Guanzhong dairy goats (2.5–3 years old, Lantian, China). GPX activity, assayed with a commercial kit (BC1195 from Solarbio), was 106.30 ÷ 133.55 U/L. 

As can be seen in Table 6, all analyzed studies reported spectrophotometric methods with the use of commercial kits. Most used kits are those for CAT and GPX, for both erythrocytes and spermatozoa, while only one manuscript reports the employment of the GSR assay in spermatozoa.

## 6. Discussion 

In the recent literature (2018–2023), different spectrophotometric, fluorimetric and chemiluminescent assay methods for ROS quantification in spermatozoa and erythrocytes of small domestic ruminants have been reported. The application of these methods requires the use of different probes and instruments, with associated disadvantages and advantages mainly related to economic aspects and the sensibility of methods. It is known that spectrophotometry is cheaper but less sensitive, while chemiluminescence is the most sensitive and expensive method. However, in the last five years, the most widely used system for ROS assay is the fluorometric method, probably because it allows to assay several ROS at the same time, using different probes, and resulting in a favorable cost-effective method. As can be seen in Table 3, the most widely used probe for fluorescence assays, in particular flow cytometry, is DCFH-DA and its derivatives. They present several advantages over other probes, such as ease of use, cell permeability, high sensitivity to changes of the redox state of a cell, and cost-effectiveness. Moreover, this kind of probes are suitable to follow changes in ROS over time [99], and so, they are suitable for kinetic studies. On the other hand, the major disadvantage of such probe is that it allows to assay only hydrogen peroxide and not others ROS. This limit has been overcome by several authors combining it with the use of other specific probes, such as DHE, that allows to identify the superoxide radicals [80,90,100]. In the last five years, many authors associate the DCFH-DA probe and its derivatives with PI [80,83,98,99,101]; this is important to avoid false positives, especially in cyto-fluorimetry by minimizing autofluorescence problems [120], and to evaluate ROS production only in living cells that are PI negative rather than in the whole cell population [121].

In recent years (2018–2023), only two authors have performed ROS assay in spermatozoa using CellRox by fluorimetry. CellRox is a cell-permeant dye, that has the disadvantage of being weakly fluorescent also in the reduced state (while exhibits bright green photostable fluorescence upon oxidation by ROS). So, when it is used for ROS detection in florescence, it could cause a false positive. 

Moreover, the use of a flow cytometer does not allow to see the distribution of fluorescence signals on/in the cell. Therefore, the results of flow cytometry assays might be misleading. Different authors developed strategies to minimize false positive signals and artifacts of flow cytometry. For example, by accounting for DNA content, changes in cell morphology, dye uptake and retention, and target specific dye activation in cells [122] or using another dye (for example, CellRox with hydroethidine) to make a distinction between intracellular superoxide and superoxide in dead cells and surrounding area [123].

Given this information, the choice of using one method over the other depends on the equipment available in the laboratories and the types of ROS to be assayed. 

Concerning the determination of intracellular antioxidant enzymes, the recent literature reports only spectrophotometric methods with absorbance measurements and many of them use commercial kits. It is important to point out that in domestic ruminant species both erythrocytes and spermatozoa can be obtained in greater quantities than in other animal species. Therefore, the high availability of the matrix could justify the wide use of spectrophotometric method, which is less sensitive but also less expensive.

The difficulty observed during this overview was mainly due to the low concordance between the methods, regarding treatment and sample concentration. In addition, some results obtained using the different methods were difficult to be compared because they were expressed with different units of measurement.

## 7. Conclusions

The advantage of quantifying ROS and antioxidant enzyme activity in the two analyzed cell types is that the obtained data could find application in studying the welfare and health status of small domestic ruminants. 

OS could be indicative of a loss in animal production because, at cellular level, altered sperm metabolism and the incidence of immature, abnormal, or dead spermatozoa are associated with ROS overproduction [71]. Spermatozoa damage or their low quality affects two important aspects: the first is related to the loss of total production, and the second is related to genetic improvement, and thus to the non-transmission of hereditary traits useful to production. Therefore, mitigating OS could improve productivity indicators in farm animals. In addition, OS effects in the erythrocytes of farm animals causes immune activation, eryptosis, and modifications at the cell membrane level that could affect health and welfare of animals [18]. Early detection of OS could be a valuable tool to avoid economic losses in the livestock sector. This review reports a compendium of the most recent methods used for the monitoring oxidative stress in the cells, providing a useful tool to researchers and farm operators for monitoring livestock production. 

## Figures and Tables

**Figure 1 animals-13-02300-f001:**
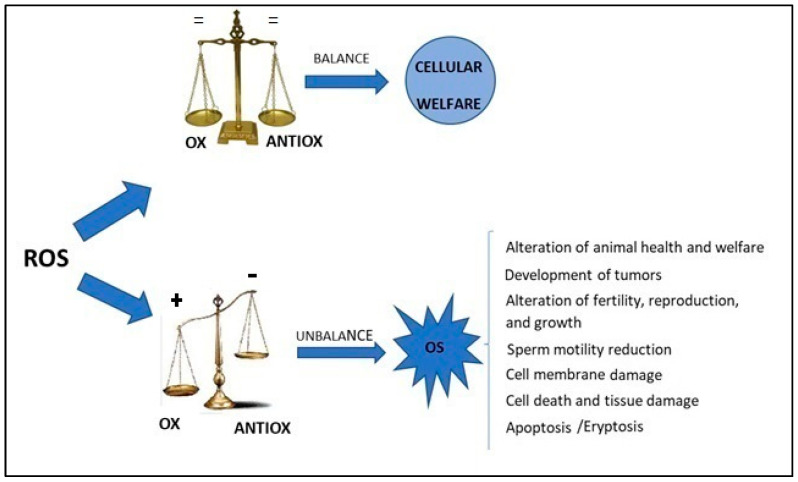
The imbalance between antioxidants and oxidants in the cell and consequences of oxidative stress. ROS = reactive oxygen species, OX = oxidant, ANTIOX = antioxidant, OS = oxidative Stress.

**Figure 2 animals-13-02300-f002:**
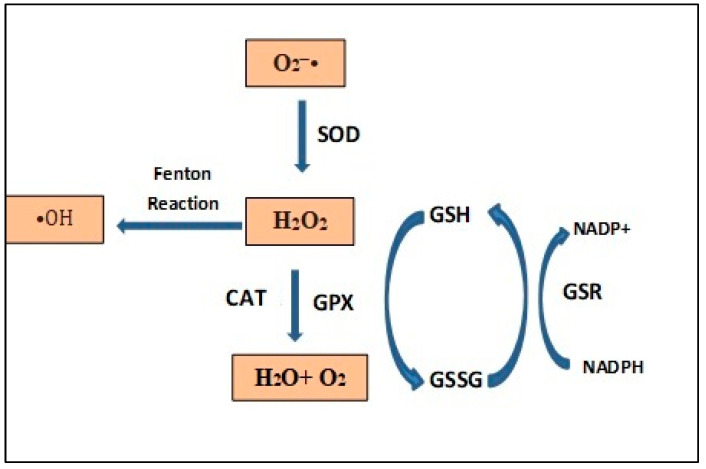
Oxidative state of the cell with the main ROS and the system of enzymatic antioxidants. Superoxide dismutase (SOD), catalase (CAT), glutathione peroxidase (GPX), glutathione reductase (GSR), oxidated glutathione (GSSG) and reduced glutathione (GSH).

**Table 1 animals-13-02300-t001:** The principal ROS in animals’ cells and tissue.

ROS	Specie	Catalyzed Reaction	Production	Refs.
Superoxide Radical	O_2_^−•^	NADPH + 2O_2_ → NADP^+^ + 2 O_2_^−•^ + H^+^	Oxygen metabolism	[19]
Hydrogen Peroxide	H_2_O_2_	2 O_2_^−•^+ 2H^+^ → H_2_O_2_ + O_2_	- Dismutation of O_2_^−•^ - Different oxidases - Catabolic reactions	[20,21,22]
Hydroxyl Radical	^•^OH	H_3_O^+^ + e^−^ → ^•^OH + H_2_ H_2_O_2_ → ^•^OH	- Aconitase reactions - Fenton reaction	[20,23]
Hydroperoxyl Radical	HOO^•^	O_2_^−^ + H_2_O → HOO^•^ + OH^−^	The protonated form of O_2_^−•^	[20]
Peroxyl Radicals	ROO^•^	R^•^ + O_2_ → ROO^•^	Polyunsaturated fatty acid metabolism	[24]

**Table 2 animals-13-02300-t002:** The major physiological antioxidants in animals’ cells and tissues.

Enzymatic Antioxidant	Function	Catalyzed Reaction	Refs.
Superoxide Dismutase (SOD)	Detoxification O_2_^−•^	2O_2_^−•^ + 2H^+^ → O_2_ + H_2_O_2_	[26,27,28]
Catalase (CAT)	Detoxification H_2_O_2_	2H_2_O_2_ → 2H_2_O + O_2_	[28,29]
Glutathione peroxidase (GPX)	Detoxification H_2_O_2_	2H_2_O_2_ + GSH → 2H_2_O+ O_2_ + GSSG	[28]
Glutathione reductase (GSR)	Restoration of the GSH by reducing the GSSG	GSSG + NADPH → GSH +NADP^+^	[28]

**Table 6 animals-13-02300-t006:** Analytical methods for CAT, GPX, and GSR detection in erythrocytes and spermatozoa of small domestic ruminants.

	Specie	Method	Lysis	Concentration	Refs.
ERYTHROCYTES	Sheep	Commercial kit for GPX and CAT. (Nanjing Jiancheng Institute of Biotech., China)	By ultrapure water	CAT: 1 ÷ 5.5 U/mg protein GPX: 10 ÷ 65 U/mg protein	[49]
SPERMATOZOA	Buck	Commercial kits: GSR: Glutathione reductase kit (Randox Lab., UK) GPX: Ransel kit (Randox Lab., UK) CAT: Catalase assay kit (Cayman Chemical Co., MI)	Freeze–thaw in cooled distilled water	GSR: 2000 ÷ 3500 U/g protein GPX: 3200 ÷ 4000 U/g protein CAT: 300 ÷ 750 nmol/min/g protein	[116]
	Goat	Commercial kit for GPX and CAT (Nanjing Jiancheng Bioeng. Institute, China)	by 0.1% Triton X-100	CAT: 1.8 ÷ 3.10 U/mL GPX: 18 ÷ 33 U/mL	[112]
		Commercial kit for GPX and CAT (Nanjing Jiancheng Bioeng. Institute, China)	by 0.1% Triton X-100	GPX: 20 ÷ 80 U/mg protein CAT: 2.5 ÷ 3 U/mg protein	[113]
		Commercial kit BC1195 for GPX (Solarbio)	N.S.	GPX: 106.3 ÷ 133.5 U/L	[104]
	Ram	Commercial kits A005–1-1 (for GPX) A007–1-1 (for CAT) (Nanjing Jiancheng, China)	Sonication in 0.9% NaCl and 1% Triton X-100	GPX: 10 ÷ 16 U/mg protein CAT:0.12 ÷ 0.2 U/mgprotein	[119]

CAT = catalase, GPX = glutathione peroxidase, GSR = glutathione reductase, N.S. = not specified.

## Data Availability

Not applicable.

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
