# Peer review of "An Overview on Assay Methods to Quantify ROS and Enzymatic Antioxidants in Erythrocytes and Spermatozoa of Small Domestic Ruminants"

_animals, 2023, doi:10.3390/ani13142300_

Round 1

Reviewer 1 Report

The manuscript titled "An Overview on Assay Methods to Quantify Ros and Enzymatic Antioxidants in Erythrocytes and Spermatozoa of Small Domestic Ruminants" is really an original and interesting topic where the authors recopile the last advances about assay methods related to ROS and Enzymatic Antioxidants in Erythrocytes and Spermatozoa in small ruminants. I think that it is really interesting the relation stablished between ROS and health/welfare, and in future papers I encourage the authors to follow this research line exploring the effects in other species too.

As I have said it is really interesting but there are some minor aspects to review before its final publication.

Line 48. "pollution" . what do you mean? environmental pollution? please specify

Line 48. "environmental contaminants" please include  some examples of contaminants between parenthesis

Figure 1.  Wouldn't the developement of reproductive problems include impairment of fertility, reproduction, sperm motility reduction...?please review the list of the health and wellfare disturbances, and avoid duplication.  

Line 221. figure 2 or Figure 2?

Line 296. ........38.5â—¦C Please correct 38.5ºC

Line 300......"Raza" instead of "Rasa" or "Aragonesa Breed" instead of "Rasa Aragonesa"

Line 422-426. There seems to be a shading in the paragraph, please review and remove it

Tables 4, 5 and 6. There seems to be a shading, please check if it is correct or if it should be removed

Lines 472-543. A gray shading appears between lines 472-543, check if it should be left

Conclusion. could oxidative stress be indicative of a loss in animal production? If this OS is detectected early, could measures be implemented to avoid economic losses in the sector.

Icono de Validado por la comunidad

Author Response

Dear Reviewer

Thank you very much for your efforts in the revision of our manuscript. Your comments and advice substantially improved the quality of the manuscript. All added and correct parts are in red in the resubmitted text.

Reviewer 1

The manuscript titled "An Overview on Assay Methods to Quantify Ros and Enzymatic Antioxidants in Erythrocytes and Spermatozoa of Small Domestic Ruminants" is really an original and interesting topic where the authors recompile the last advances about assay methods related to ROS and Enzymatic Antioxidants in Erythrocytes and Spermatozoa in small ruminants. I think that it is really interesting the relation stablished between ROS and health/welfare, and in future papers I encourage the authors to follow this research line exploring the effects in other species too. As I have said it is really interesting but there are some minor aspects to review before its final publication.

Q1: Line 48. "pollution". what do you mean? environmental pollution? please specify 

R1: Thanks, as suggested, “environmental” has been added in the text

Q2: Line 48. "environmental contaminants" please include some examples of contaminants between parenthesis

R2: As suggested, several examples of contaminants between parenthesis (Bisphenols, Pesticides, Metals and Metalloids) have been added.

Q3: Figure 1.  Wouldn't the developement of reproductive problems include impairment of fertility, reproduction, sperm motility reduction...? please review the list of the health and wellfare disturbances, and avoid duplication.  

R3: Thanks, as suggested the list in Figure 1 has been modified.

Q4: Line 221. figure 2 or Figure 2?

R4: As suggested, the word has been corrected.

Q5: Line 296. ........38.5â—¦C Please correct 38.5ºC

R5: As suggested, °C has been corrected.

Q6: Line 300......"Raza" instead of "Rasa" or "Aragonesa Breed" instead of "Rasa Aragonesa"

R6: As suggested, Raza has been corrected.

Q7: Line 422-426. There seems to be a shading in the paragraph, please review and remove it

R7: As suggested, we have removed the shading in the paragraph indicated.

Q8: Tables 4, 5 and 6. There seems to be a shading, please check if it is correct or if it should be removed

R8: As suggested, we have removed the shading in the tables indicated.

Q9: Lines 472-543. A gray shading appears between lines 472-543, check if it should be left

R9: As suggested, we have removed the shading in the lines indicated.

Q10: Conclusion. could oxidative stress be indicative of a loss in animal production? If this OS is detected early, could measures be implemented to avoid economic losses in the sector.

R10: yes, the OS could be indicative of a loss in animal production because, at cellular level, altered sperm metabolism and the incidence of immature, abnormal, or dead spermatozoa are associated with ROS overproduction [71]. Spermatozoa damage or their low quality, affects two important aspects: the first is related to the loss of total production, and the second is related to genetic improvement, and thus to the non-transmission of hereditary traits useful to production. Therefore, mitigating OS could improve productivity indicators in farm animals. In addition, OS effects in the erythrocytes of farm animals causes immune activation, eryptosis, and modifications at the cell membrane level that could affect health and welfare of animals [18]. Early detection of OS could be a valuable tool to avoid economic losses in the livestock sector. We have added this sentence in the conclusion of this review.

Reviewer 2 Report

Dear Authors, thank you for your interesting review. The presented review covers an important topic of oxidative stress and antioxidants, and methods of its evaluation in small ruminants. In general, I have to say such a review is definitely a good contribution to the field. However, I have a few major and minor critiques of the manuscript in its current form:

Major:

1) The chosen timeframe (2020-2023) must be extended to 2018-2023 (to include the last 5 years, at least);

2) The review should be focused either on red cells or on sperm cells. I strongly recommend focusing the review on spermatozoa;

3) The section "Discussion" must be included. In this section, the advantages and (especially) disadvantages of reviewed assays must be discussed. Your review must be a guide for the researchers to make a correct choice of methodology for assaying oxidative stress or antiox activity. 

4) For particular flow cytometry-based assays (for example, the CellRox) for the detection of oxidative stress in sperm cells, the danger of obtaining false-positive results must be discussed (see for example Davila et al. (2015) doi:10.1371/journal.pone.0138777). This is important because, with the use of flow cytometry-based assays, it is not possible to see the distribution of fluorescence signals on/in the cell. Therefore, the results of such assays might be misleading;

5) Please, include in the manuscript the passage about heat stress and oxidative stress and their relationship;

Minor:

1) In Table 3, the Concentration ROS should be given in comparable units.

Once again I thank the authors for their work. I hope my criticism will be helpful and will increase the quality of the manuscript.

Author Response

Dear Reviewer

Thank you very much for your efforts in the revision of our manuscript. Your comments and advice substantially improved the quality of the manuscript. All added and correct parts are in red in the resubmitted text.

Reviewer 2

Dear Authors, thank you for your interesting review. The presented review covers an important topic of oxidative stress and antioxidants, and methods of its evaluation in small ruminants. In general, I have to say such a review is definitely a good contribution to the field. However, I have a few major and minor critiques of the manuscript in its current form:

Major:

Q1) The chosen timeframe (2020-2023) must be extended to 2018-2023 (to include the last 5 years, at least);

R1) As suggested, the chosen timeframe to 2018-2023 has been extended.

Q2) The review should be focused either on red cells or on sperm cells. I strongly recommend focusing the review on spermatozoa;

R2) Thank you for your suggestion.

At the beginning, we thought to investigate analytical methods for two different cells in two separate reviews, but the following considerations led us to decide to keep both:

1) Generally, assay methods for ROS and antioxidant enzymes are the same for both cells

2) Both cells are unable to activate transcription processes to increase their antioxidant defenses, because of their intrinsic nature: erythrocyte is an enucleated cell in mammalian and the spermatozoon is a haploid cell.

3) Spermatozoa and erythrocytes are both highly susceptible to the deleterious ROS effects, due to the large amount of unsaturated fatty acids present in their cell membranes.

4) Our research group works on both cell types, and we preferred to merge spermatozoa and erythrocytes because the methods to quantify ROS are the same; we didn't want to risk writing two reviews describing the same methods.

Furthermore, much more time should be necessary to rework the review by focusing on only one cell type. If the reviewer feels it is absolutely necessary, I might try to ask for an extension of the available time.

Q3) The section "Discussion" must be included. In this section, the advantages and (especially) disadvantages of reviewed assays must be discussed. Your review must be a guide for the researchers to make a correct choice of methodology for assaying oxidative stress or antiox activity.

R3) As request, the section “Discussion” has been included in the text, and the advantages and disadvantages of reviewed assays have been discussed.

Q4) For particular flow cytometry-based assays (for example, the CellRox) for the detection of oxidative stress in sperm cells, the danger of obtaining false-positive results must be discussed (see for example Davila et al. (2015) doi:10.1371/journal.pone.0138777). This is important because, with the use of flow cytometry-based assays, it is not possible to see the distribution of fluorescence signals on/in the cell. Therefore, the results of such assays might be misleading;

R4) Thank you for your suggestion. The limit of flow cytometry-based assays (for example, the CellRox) has been added in the Discussions section.

Q5) Please, include in the manuscript the passage about heat stress and oxidative stress and their relationship.

R5) We thank the reviewer for pointing this out. This has been added in the text (INTRODUCTION). Different authors indicated a relationship between heat stress (HS) and OS, due to similar genes expressed after heat or oxidant agents’ exposure [34,35]. Indeed, HS was also proven to increase antioxidant enzyme activities (SOD, CAT and GPX) as consequence of increased ROS levels after heat exposure [36].

 HS was proven to be responsible of inducing oxidative stress in sheep [36–38]. The increased environmental temperature and humidity, that occur during summer, can compromise the animal production in livestock industries [39].

Slimen et al. [34] reported that HS affects the oxidative status of blood sheep leading to an overproduction of transition metal ions, that determine electron donations to oxygen, forming of superoxide anion and/or hydrogen peroxide.

Furthermore, Shahat et al. [40] demonstrated that mild HS induces deleterious effects on animal health and reproduction, causing total and/or progressive reduction of sperm motility and acrosome integrity of fresh spermatozoa. These effects can be mitigated by slow release of substances with antioxidant effect as melatonin, taken before of HS exposure. 

Minor:

Q1) In Table 3, the Concentration ROS should be given in comparable units.

R1) this issue had already been pointed out with the phrase “some results obtained from the different methods were difficult to be compared because they were expressed with different units of measurement” which now appears at the end of the Discussion.

It would be very useful to have a table with the same units of measurement for ROS assayed in spermatozoa: most are expressed in % probe-positive cells out of total spermatozoa counted, or in Rfu, one in mgH2O2/108 spermatozoa and one in U/mL (this is a unit of measurement obtained after the assay of ROS with a commercial kit of which no information is given). So, with great regret, we cannot create a table using a single unit because we do not have all the elements to make this conversion.

Once again I thank the authors for their work. I hope my criticism will be helpful and will increase the quality of the manuscript.

Round 2

Reviewer 2 Report

Dear Authors, I thank you very much for your reformulated manuscript. Currently, I only have a comment on the references list: please, check and correct the list (there are several citation styles used together; please correct according to journal standards).

Thank you.